# Variation in Juvenile Long Bone Properties as a Function of Age: Mechanical and Compositional Characterization

**DOI:** 10.3390/ma16041637

**Published:** 2023-02-16

**Authors:** Claudia Vázquez Sanz, Ignacio Victoria Rodríguez, Francisco Forriol, Elena Tejado, Francisco J. Lopez-Valdes

**Affiliations:** 1Instituto de Investigación Tecnológica (IIT), ICAI School of Engineering, Universidad Pontificia Comillas, 28015 Madrid, Spain; 2Departamento de Ciencia de Materiales-CIME, Universidad Politécnica de Madrid, 28015 Madrid, Spain

**Keywords:** cortical bone, aging, elastic modulus, nanoindentation, composition

## Abstract

Bone is a heterogeneous, hierarchical biocomposite material made of an organic matrix filled with a mineral component, which plays an important role in bone strength. Although the effect of the mineral/matrix ratio on the mechanical properties of bone during aging has been intensively investigated, the relationship between the mechanical properties and the chemical composition of bone with age requires additional research in juvenile individuals. In this study, bone coupons from bovine and ovine animal species were machined from cortical areas of long bones to quantify whether the variation in mechanical properties at different stages of development is related to the change in the composition of bone tissue. An energy-dispersive X-ray detector (EDX) attached to a scanning electron microscope (SEM) was used to perform a compositional analysis of the tissue. In addition, nanoindentation analyses were carried out to address how the elastic modulus changed with age. Nonparametric statistical analyses found significant differences (*p* < 0.05) in Ca content and elastic modulus between species, but no differences were found within each species with development. A multiple linear regression model found that the elastic modulus was significantly related to the decrease in P and C in the samples, to the animal species (larger in bovine), and development, although not linearly. This model also found an interaction between Ca and development that could explain the lack of significance of the relationship between the elastic modulus and development in the univariate models.

## 1. Introduction

Bone is a heterogeneous, anisotropic, natural composite material consisting mainly of collagen fibrils (30–40 weight %), a mineral phase (50–60 weight %), and water (10–20 weight %). Bone structure can support physiological loads and is able to partially tune its structural mechanical behavior through remodeling. Multiple factors contribute to the structural response of bone tissue: total bone mass, bone geometry, microdamage and microstructural discontinuities (such as microporosity), and the mechanical properties of its constituents. Changes in the proportions of the constituents may have significant effects on bone fragility. Among them, the mineral phase, with a greater elastic modulus, provides rigidity to the bone and makes it behave anisotropically due to the uneven distribution of crystal minerals [1].

The collagen phase also plays an important role in the mechanical behavior of bone tissue: while the mineral component contributes to bone stiffness, the collagen phase is related to the toughness of bone tissue [2].

These constituents are not constant during the development and aging of individuals. In humans, the maximum bone mass is reached in the third or fourth decade of life, and then both the quantity and the quality of bone tissue in the human skeleton begin to deteriorate [3]. Changes in metabolism that occur with age modify the structure of bone, affecting its strength and toughness [1,3].

Cortical bone consists mainly of concentric bone lamellae arranged around blood vessels that form osteons and interstitial areas. It has a porosity of around 5–15%, mainly oriented in the osteon direction [4]. From a mechanical perspective, porosity has been confirmed to be one of the most significant determinants of mechanical properties in bone because it reduces the load-bearing area, effectively causing more stress on the material [3,4]. For example, age-related increases in collagen porosity can directly lead to decreases in collagen phase stiffness and strength [1]. Therefore, osteoporosis, which involves loss of bone quantity due to resorption, directly deteriorates the mechanical properties of bone tissue with age [3].

Although the change in bone tissue composition and mechanical properties is well understood in the elderly, there is a paucity of experimental data on the evolution of these characteristics in the early stages of development [5,6]. Recent studies have reported an increase in bone modulus with young age [5,7,8]; however, these studies emphasize the need to collect more samples to provide stronger findings. In the case of human tissue, this lack of experimental data may be related to the difficulties associated with obtaining tissue donors at an early age [9]. Another practical consideration relevant to any species is that obtaining and mechanically testing tissue samples from juvenile and young specimens could be challenging due to their inherent smaller size. However, the characterization of the composition and mechanical properties of bone tissue in younger individuals is key in the development of models (such as finite element models of the human body) that can be used to approximate the mechanical response of children [10]. Traditionally, pediatric mechanical properties of children were obtained by scaling the properties measured in experiments carried out with adult tissue, but several studies have shown the limitations of this approach [11,12].

Moreover, the rapid evolution in biomaterial performance in the last decade has caused an increasing need to classify their functionalities, especially regarding mechanical properties [13]. In this sense, the development of personalized tissue engineering scaffolds will also benefit from the data collected in the characterization of juvenile bone tissue. Properties such as the elastic modulus and its variation with mineral content and porosity are key to the development of the aforementioned bone scaffolds [14].

To assist in the analysis of how bone composition and the elastic modulus change during early development, this study includes the results of bone tissue samples from two mammal species (bovine and ovine) obtained at different stages of juvenile development. 

The rationale behind using individuals from two species is to identify whether similar trends in the change in mechanical properties with development can be identified in the two species. If so, this trend could also be applied to improve the scaling methods mentioned above that are applied to adult human tissue. 

Samples were extracted from cortical bone tissue from large bones and from individuals at three different stages. The composition of the tissue was characterized by energy-dispersive X-ray analysis (EDX) technique using a scanning electron microscope (SEM), while the material properties were obtained from nanoindentation experiments. 

Therefore, the objective of this research project was to quantify the change in elastic modulus and composition at different stages of bone tissue development with a focus on juvenile individuals.

## 2. Materials and Methods

Long bones (tibia and femur) of two mammal species (bovine and ovine) at different stages of development were obtained from local providers (Mercamadrid and Carrefour in Madrid area). Bovine samples were machined from the femur, while both the femur and the tibia were used for the ovine specimens. Nutrition was used as a proxy for age, establishing three age groups: breastfed, transitioning to solid food (SFT), and only solid food (mature). Breastfed samples include cows up to 12 months of age and sheep up to 4 months; solid-food transitioning specimens included those individuals who started to graze, including bovine subjects between 12 and 18 months and ovine subjects within the 4–12 month range; mature individuals included cows older than 18 months and sheep older than 12 months.

After cleaning the bone from soft tissue remains (scraping the periosteum and removing the bone marrow), the diaphyseal areas of the bone were cut into wafers perpendicular to their longitudinal axes. These wafers were then cut into 0.8 mm longitudinal sheets of cortical tissue using a circular diamond-encrusted blade saw (Remet Micromet Evolution 8502, Bologna, Italy).

Eighteen bone wafers were machined for this study. Specimens were classified into three different groups, and six samples were taken for each group (one for each age and species). The fabricated coupons were then preserved in a NaCl solution (VisClean Physiological Saline Solution, France) to keep the samples hydrated throughout the process. The tissue was preserved by freezing at −18 °C and thawed at room temperature before testing. 

Prior to testing, the surfaces of the samples were evaluated using digital microscopy imaging to detect possible defects that can lead to discontinuities in a determined area of the coupon. An Olympus DSX10-UZH high-resolution digital microscope (Tokyo, Japan) equipped with a tilt frame system and a telecentric lens was used to perform the surface inspection. 

In addition, the coupons were examined to address how the phase composition of the mineral bone varied with age using a scanning electron microscope (SEM, ZEISS Auriga Gemini, Erfurt, Germany). 

This microscope provides high-resolution and high-magnification images of the material’s surface as a result of the interaction of the electrons with the sample. 

The microscope is equipped with secondary electron detectors (SE), backscattered electron detectors (BSE), and X-ray dispersion detectors, which allow energy-dispersive X-ray analysis (EDX) to identify the elemental composition of materials. This method involves the analysis of X-rays with characteristic energies of the atoms of the specimen, emitted when an incident electron hits the sample surface [15]. EDX allows obtaining both qualitative and quantitative information on the composition of the sample since the energy of each detected X-ray is characteristic of the element that emitted it. For the compositional analysis, QUANTAX software (Bruker AXS) was used. The aforementioned software allows the configuration of a point cloud matrix throughout the entire surface of the samples in order to obtain average representative results of the composition of the specimens.

Observation of samples in the SEM required additional tissue preparation. A metalizing process was applied to the samples right before they were analyzed to ensure proper conductivity of the samples. To this end, they were mounted on conductive metallic holders using conductive carbon tape. After that, a 12.67 nm carbon layer was deposited in each of the samples using a Leica EM ACE600 (Wetzlar, Germany) metallizer. The metalizing vacuum pressure was 10^−4^ mBar, and 26 pulses were required to deposit the adequate thickness of the conductive layer. The amount of carbon deposited on each sample was controlled and was homogeneous between samples.

Lastly, the elastic modulus of each sample was characterized by instrumented indentation analysis. Nanoindentors can continuously measure force and displacement as the indentation is performed, avoiding the limitations of optically measuring the imprints.

The technique uses a diamond indenter to load and unload a material specimen, leaving a permanent indent on the surface. Data were obtained from a complete loading and unloading cycle and analyzed according to the model proposed for the deformation of an elastic half-space by a rigid axisymmetric punch previously derived by Sneddon [16]. The mathematical solution was adapted for the geometry of the nanoindenter by Oliver and Pharr [17]. The stiffness *S* is related to the reduced modulus *E_r_* by
(1)$$S=\frac{dP}{dh}=\beta \;\frac{2}{\surd \pi }E_{r\;}\sqrt{A}$$
where *A* is the projected contact area (a function of depth, *h*) and *β* is an empirical shape factor. The reduced modulus accounts for using a non-rigid indenter and is determined by the following equation:(2)$$\frac{1}{E_{r}}=\frac{\left(1-\nu_{s}^{2}\right)}{E_{s}}+\frac{\left(1-\nu_{i}^{2}\right)}{E_{i}}$$
where *E_s_* and *ν_s_* are the sample modulus and Poisson’s ratio, respectively, and *E_i_* and *ν_i_* are the indenter modulus and Poisson’s ratio, respectively.

Nanoindentation tests were conducted using an XP Nanoindenter (MTS Nano Instruments, Oak Ridge, TN, USA) equipped with a Berkovich diamond tip. Bone coupons were cut into smaller pieces using a water-cooled diamond wire saw (Well, Walter Ebner, Germany) to later embed them in epoxy resin (Mecaprex MA2+, Presi, France). Once the resin was cured, the mounted specimens were wet-polished, applying gentle pressure using an ABRAMIN Struers (Germany) polisher. Different silicon carbide sandpaper rugosities (400, 600, and 1200 grit) were used until the desired surface finish was achieved. Polishing was continued on napless cloths with 3 µm and 1 µm diluted diamond pastes. Finally, the samples were washed with water and observed under the optical microscope to check whether the desired surface finish was achieved. Each sample was indented with a 10 × 5 array, and a 25 µm distance was set between each indentation to avoid interaction between each indent. The load control tests were carried out at up to 5 mN, resulting in a penetration depth of 500 nm. In this study, a 0.3 Poisson modulus previously suggested in the literature ([13,14]) was considered, which is consistent with recently reported values for human femoral cortical bone (0.28).

### Statistical Analysis

One-way variance analyses were performed with the data gathered from compositional and nanoindentation tests. First, the effect of species (bovine and ovine) on composition and elastic modulus was analyzed. After checking the homogeneity of the variance, Mann–Whitney U tests were performed to compare the statistical significance of the two species in calcium content and elastic modulus. 

Furthermore, Kruskal–Wallis tests were carried out to determine the influence of development on Ca content and elastic modulus. This test represents a nonparametric alternative to one-way analysis of variance. 

Finally, a multiple regression analysis was performed in which all predictors (element content, age, and species) were regressed against the elastic modulus to determine how much of the variation in the elastic modulus is explained by each predictor [18]. 

The significance level for all statistical tests was set at *p*-value < 0.05.

## 3. Results

As mentioned above, 18 juvenile bone samples were characterized in this study. Bone samples were named according to the developmental stage of feeding of the animals and the species to which they belong. V1 and O1 were used to refer to breastfed specimens (bovine and ovine, respectively), V2 and O2 referred to specimens transitioning to solid food, and V3 and O3 were used to identify the mature specimens.

An after-dot second digit was added to indicate the specimen number within a specific age group and species. For example, sample V1.2 refers to the second specimen within the breastfed bovine category.

### 3.1. Compositional Tests

The result of the EDX composition analysis is a spectrum in which the size of the peaks is related to the proportional content of each specific element, as shown in Figure 1. Compositional analyses revealed a high presence of calcium and phosphorus, although other elements were also found in smaller proportions, such as sulfur, magnesium, sodium, and carbonate salts (Figure 1, Table 1).

The Ca content increased with age in juvenile bovine and ovine bone tissue. In the mature ovine group, the percentage of Ca obtained was slightly lower compared to the solid-food transitioning specimens. However, its high standard deviation (±2.91) could explain this difference. 

The values of the bovine samples ranged between 34% and 41% of the Ca content, while for the ovine samples, the results were between 33% and 36% (for breastfed and mature samples, respectively) as depicted in Figure 2 and Table 2.

### 3.2. Nanoindentation Analysis

Nanoindentation testing was used to obtain local mechanical properties in bone samples. The imprints left by the nanoindenter after testing the bone specimens are shown in Figure 3 for illustration purposes.

The load and displacement curves depicted in Figure 4 represent the typical response of two samples (V1.3 and O1.4) during the indentation tests. These two samples were chosen just for illustration purposes, but all samples tested showed similar behavior. The elastic modulus was calculated from the slope of the initial segment (50 %) of the loading curve. The results obtained for each sample with their respective standard deviations are shown in Table 3. In addition, Table 4 contains the means of the elastic modulus values for each age group with their respective deviations. As mentioned above, 50 indents (matrices of 5 × 10) were made on each of the bone samples to obtain representative values of the overall behavior of each material.

Figure 5 shows a clear trend of increasing the elastic modulus with age for juvenile specimens for either species. The mean numerical values of the elastic modulus for breastfed bovine bones are 18.7 ± 2 GPa, 21.1 ± 2 GPa for solid-food transitioning animals (SFT), and 21.5 ± 2.6 GPa for mature specimens. The average values of the modules for the ovine specimens were 17.4 ± 2.8 GPa for breastfed animals, 17± 3 GPa for solid-food transitioning, and 18.5 ± 4 GPa for mature bones.

### 3.3. Statistical Analysis

The results of the one-way variance analysis (*Mann–Whitney U*) indicated that the Ca content had a significant variation between species (*p* < 0.05), and the same happened when evaluating changes in elastic modulus between the two species (*p* < 0.05). The statistical test confirmed the previous results, which showed values of both Ca content and elastic modulus that were lower for the ovine specimens.

This is probably related to the higher porosity content that is typical of ovine cortical bone, which contributes to reducing the magnitude of these mechanical properties.

The nonparametric *Kruskal–Wallis* tests showed that there were no significant differences with age when evaluating the Ca content and the elastic modulus (*p* > 0.05). From the experimental campaign, it was suggested that these parameters increased with increasing age, but the reduced sample size did not allow enough statistical power to prove this relationship. 

The dependence of the elastic modulus on the percentages of calcium, phosphorus, and carbon mass, age, and species was investigated by multiple linear regression analysis (Table 5). Interaction terms between Ca content and age and Ca content and species were also included, as these variables improved the R^2^ coefficient of regression. Both the content of P and C were significantly (*p*-value < 0.05) associated with a decrease in bone elasticity modulus. 

Individuals in the mature group exhibited a larger modulus compared to the breastfed group. The solid-food transitioning group had a more complex behavior, as the interaction between Ca content and belonging to this group was significant. This interaction controls that for specimens in this group with a higher Ca content, the modulus increased, while if the Ca content was below a certain threshold, the modulus would be reduced regardless of the more advanced stage of development. In other words, while the influence of being in the mature group in exhibiting a larger elastic modulus is independent of the Ca content, there is an association between age and Ca content that dictates the resulting modulus of the tissue. The ovine subjects exhibited a significantly lower value of the modulus of elasticity compared to bovine specimens, as identified in the previous test.

## 4. Discussion

The results of this study show that development during the juvenile stages can have significant effects on the elastic modulus and chemical composition of the samples analyzed. It was found that belonging to a different animal species was significantly related to changes in the aforementioned characteristics.

A compositional analysis was performed on 18 long bone samples to identify the elemental composition of bone tissue and to assess whether the changes in the former were related to variations in mechanical properties. The inorganic component of bone is primarily crystalline hydroxyapatite, [Ca_3_(PO_4_)_2_]3Ca(OH)_2_, while the organic components of bone include more than 30 proteins, type I collagen being the most abundant. The inorganic part may contain impurities, and the most common is carbonate in place of the phosphate groups. 

Other known substitutions may include potassium, magnesium, strontium, and sodium in place of calcium ions, and chloride and fluoride in place of hydroxyl groups [19]. Energy-dispersive X-ray analyses detected expected compositional elements in our samples, such as calcium, phosphorous, magnesium, or carbonate salts [20]. These chemical elements were incorporated during animal bone development, although the Ca/P ratio may vary according to nutritional conditions.

Although nonparametric variance tests revealed that there were no statistically significant variations in Ca content within the different age groups, the compositional analysis revealed that the percentage of calcium mass, and therefore mineralization and crystallization, increased with age for juvenile tissue in both species. Furthermore, statistical tests demonstrated that calcification varied widely between species: bovine samples showed higher Ca contents compared to ovine samples, most likely due to the increased porosity of the latter. Bone development generally experiences remarkable changes regarding the shape and size of the tissue. Bone growth is ruled by continuous Ca homeostatic remodeling, which repairs fractures and gives shape to the bone during the early stages [21]. The percentages of calcium and mineral in bone are different for each individual and may be related to genetic factors [22]. However, the Ca and mineral quantities should remain constant, and this is achieved with balanced nutrition. There are references in the literature that show that Ca percentages remain almost constant in juvenile subjects right before puberty, and this could be one of the reasons why nonparametric variance tests did not find statistical significance within different age groups [23].

Sulfur is a constituent of collagen that can be found in tissues that contain large amounts of protein. In our sample, only small proportions of sulfur were detected in a handful of bone specimens studied, despite being a constituent of collagen and therefore susceptible to be found in all samples. *Nechlich and Richards* [24] performed sulfur isotope measurements of bone collagen from archaeological sites in their study, stating that the mean amount of sulfur in mammals was 0.28 ± 0.08 %, which is consistent with the quantities detected provided by the EDX analysis. Small percentages of aluminum and even smaller concentrations of Sr were also detected in many of the samples. The presence of these minerals can be associated with the food intake of animals. With regard to aluminum, its presence can be explained by the diet of the animals, as this mineral is naturally present in large quantities in the soil. Even almost all the aluminum consumed passes through the gastrointestinal tract unabsorbed, and the amounts of dietary intake are low; around 4% of the aluminum content of the diet is retained and could partially accumulate in the bones [25,26]. Additionally, small strontium contributions were detected by EDX, but since the intensity was that low, they were not considered in the overall analysis. Sr can enter the bodies of mammals through the diet, as it is present in water and, consequently, in vegetables. Sr exhibits a chemical behavior similar to that of Ca, and it can have different protein-linking grades. Sr +2 cations can be exchanged equimolarly with Ca and therefore incorporated into hydroxyapatite crystals [27]. Contributions of sodium and chlorine were also detected, but they are likely associated with the saline solution in which the samples were stored in the refrigerator. 

The carbon content was not quantitively analyzed within groups, although it was expected that C percentages would increase with age since atoms of this element will compensate for missing Ca and P atoms [28]. It is important to note that the carbon content values are probably higher than what they should be because a homogeneous layer of this element was deposited on the samples’ surfaces before their observation on the SEM. 

In addition, carbon tape was used to attach the coupons to the microscope stubs, which could have contributed to an excess of carbon in the samples.

The percentages of magnesium mass did not follow a clear trend, either, but the variation between samples was not relevant compared to that of calcium or phosphorous.

Elastic modulus measurements obtained in the nanoindentation tests resulted in values of around 21 GPa for bovine samples and 17 GPa for ovine samples, which are consistent with other macroscopic and microscopic measurements for cortical bone. Literature values for cortical bone comprise between 17 and 20 GPa, depending on age and general tissue condition. 

Our results compare favorably to previous studies of nanoindentation performed in cortical bone, such as those reported by *Subit* et al., with values ranging between 17 GPa and 20 GPa for pediatric human femoral tissue [8]. Furthermore, in [29], elastic moduli from 19.5 GPa to 21 GPa were obtained for human femoral cortical bone. Additionally, *Casanova* et al. reported stiffness of the mice femoral cortical bone within 17 GPa and 21 GPa [30], which are also consistent with the values obtained in our study.

The experimental data showed a trend of increasing elastic modulus with age for juvenile bone specimens. These results are consistent with the existing literature [9,24]. The increase in modulus with age was applicable for both bovine and ovine specimens, as it has also been reported to increase in the case of pediatric and juvenile human bone [9]. Interestingly, our results are in agreement with the modulus of cortical tibial bone reported for pediatric (mean age: 12 years ± 3 years) and human subjects (12 GPa–20 GPa) in recent studies [6,8]. One-way variance analyses demonstrated that there were significant changes between species when evaluating the nanoindentation results, so it can be stated that the fact of belonging to a different species influences the magnitude of the elastic modulus.

*Kruskal–Wallis* tests could not find statistically significant changes in stiffness with age within species, which is consistent with some previously published studies [1,2]. Indeed, for the sample studied here, breastfed ovine specimens exhibited the same elastic modulus as the SFT ovine specimens. However, the regression model allowed us to assess that the mature age group was significantly related to an increasing elastic modulus. Unfortunately, our study is not capable of identifying whether the lack of significance of the differences between breastfed and transitioning individuals to solid food is due to a progressive development that is not possible to quantify with our data or if it is just a consequence of a reduced sample size.

In parallel, an interaction between the Ca content and the solid-food transitioning group was also identified in the regression model. This interaction affected the increase or decrease in modulus, depending on the developmental age of the sample. The aforementioned could be related to the mineral content/collagen ratio, which varies throughout bone development and confers higher or lower levels of elasticity to bone tissue. The R-squared coefficient for the regression model was 0.26, which, despite not being close to 1, is similar to the coefficients reported in published regression models relating bone content to mechanical properties [9].

To obtain more accurate results to address the effects of aging, it will be interesting to perform the tests described throughout this study on a larger population to increase the statistical power of the tests. Another limitation of this study is that while the modulus value was obtained locally at 50 locations in each sample, the amount of bone constituents was always taken as the average value of a point cloud, as there was no direct correspondence between the nanoindentation sites and the EDX analyses. 

A study by *Akkus* et al. [28], in which female rat femurs of 3 months, 8 months of middle age, and 24 months of age were analyzed using Raman spectroscopy, showed that increased mineralization and increased crystallinity were significantly correlated with decreased elastic deformation with age. *Yerramshetty* et al. [31] studied the association between mineral crystallinity and mechanical properties of human cortical bone, showing that increased tissue-level strength and stiffness were positively correlated with increased crystallinity, while ductility was reduced. Furthermore, *Currey* et al. [32] studied the effect of variations in mineral content and Young’s modulus of some mammalian mineralized tissues to discuss the implications of mineralization variation in nature. From their results, they concluded that, in general, in mineralized tissues, mineral content is associated with a higher elastic modulus but lower toughness. 

The mechanical properties of cortical bone on the macroscale are closely related to its microstructure and composition [4]. Their relationship has been extensively studied in the past [33,34,35], and the results have shown that elastic modulus, strength, and energy absorption decrease with increasing porosity or with the area fraction of osteons. For example, age-related increases in collagen porosity can directly lead to decreases in stiffness and strength. Our multiple regression analysis demonstrated that the ovine bone samples, whose porosity is high compared to the bovine species, negatively correlated with stiffness. In addition, stiffness and Ca percentages of the ovine samples were lower than the values reported for the bovine coupons. This is probably related to the higher porosity content that is typical for ovine bone tissue. From a parallel ongoing study of the ICAI School of Engineering, in which bovine and ovine bone coupons are tested under tensile conditions and subjected to fractographic analysis by SEM, it was possible to confirm that the ovine bone tissue has a higher porosity content compared to bovine samples (Figure 6). In contrast, mature age and Ca content positively influenced the elastic modulus.

## 5. Conclusions

The present work aimed to gain insight into the mechanical and compositional properties of growing cortical bone tissue to improve the knowledge of juvenile bone to cope with the existing practical difficulties of testing pediatric human tissue. The main focus of this study was to confirm that similar trends could be observed in the change in material properties in two different species of mammals. Although no statistically significant variations were reported by non-parametric statistical tests within the evaluated age groups, it was possible to confirm that the percentage of calcium mass and the elastic modulus tend to increase with age for juvenile tissue in both mammal species. Juvenile bone tissue becomes stiffer with increasing age, and calcification and mature age have a strong influence on that.

## Figures and Tables

**Figure 1 materials-16-01637-f001:**
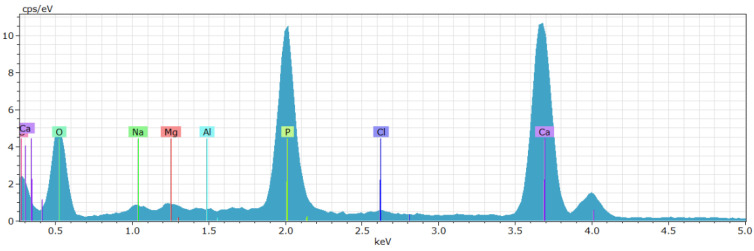
Compositional spectrum of the surface of O3.3 sample obtained using the EDX technique.

**Figure 2 materials-16-01637-f002:**
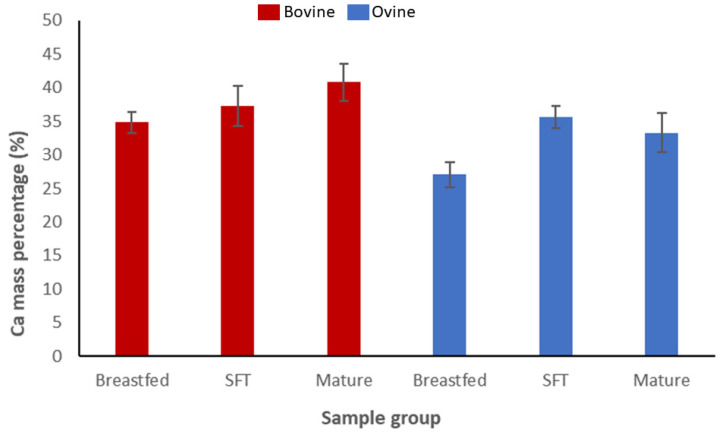
Calcium mass percentages reported with EDX analysis with their respective standard deviations. Bovine specimens are shown with red bars, whereas ovine specimens are colored blue. SFT refers to solid-food transitioning specimens.

**Figure 3 materials-16-01637-f003:**
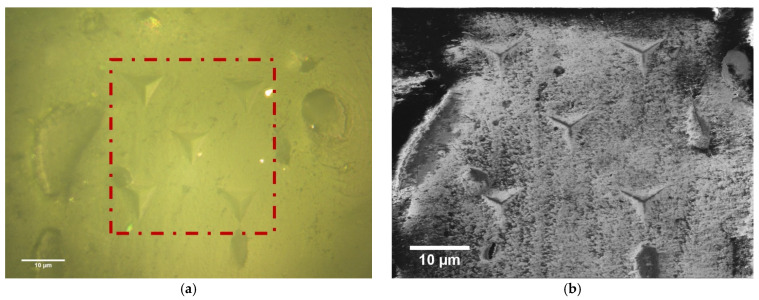
Typical imprints left by the nanoindenter for bone tissue. (**a**) Optical microscopy image of 5 isolated indentations at a depth of 2000 nm. (**b**) SEM image of five isolated indentations at a depth of 2000 nm. The distance between each indentation was 25 µm for both images.

**Figure 4 materials-16-01637-f004:**
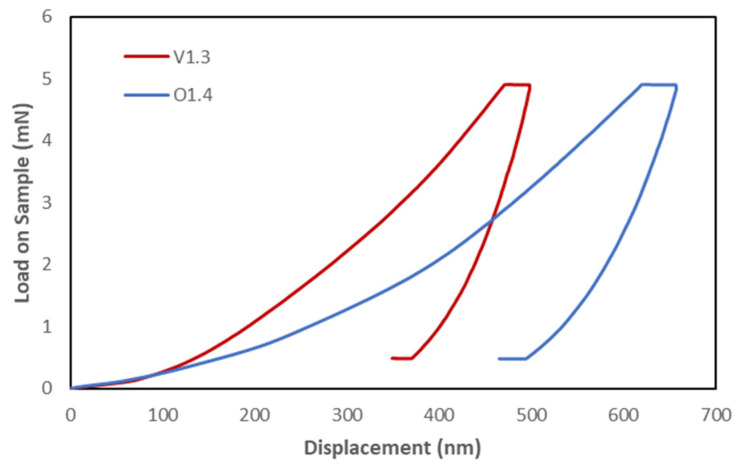
Representative load–displacement indentation curves of two bovine and ovine breastfed bone tissue samples.

**Figure 5 materials-16-01637-f005:**
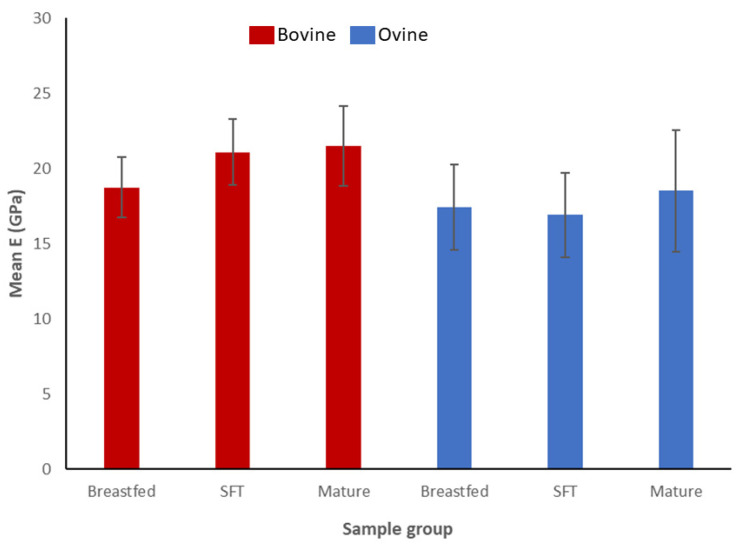
The bar chart represents the mean value of elastic modulus (E) with the respective standard deviations of the analyzed samples. Bovine specimens are shown with red bars, whereas ovine specimens are colored blue.

**Figure 6 materials-16-01637-f006:**
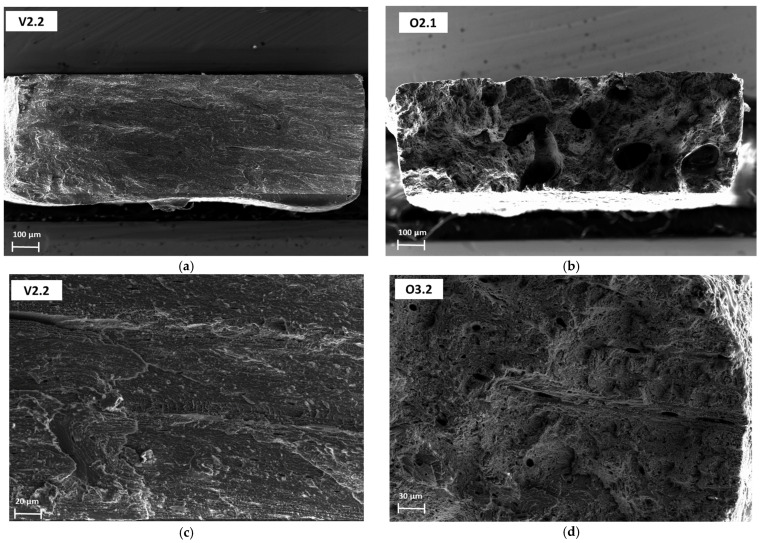
SEM fractographies of dog bone coupons cross sections that were tested under tensile loads. (**a**,**c**) show surface fractures of bovine samples, which almost do not show any signs of porosity. On the other hand, (**b**,**d**) belong to ovine bone coupons. As can be observed, these samples show high porosity content, which directly affects bone stiffness.

**Table 1 materials-16-01637-t001:** Compositional analysis results obtained using EDX. Values are displayed with respective standard deviations.

Sample ID	C	O	Na	Mg	Al	P	Cl	S	Ca
V1.1	18.36 ± 0.62	38.52 ± 0.36	0.81 ± 0.03	0.62 ± 0.01	0.05 ± 0.01	12.81 ± 0.19	0.10 ± 0.007	-	28.72 ± 0.46
V1.2	12.18 ± 1.08	30.36 ± 2.13	0.57 ± 0.08	0.35 ± 0.04	-	16.22 ± 0.50	0.27 ± 0.02	-	39.58 ± 2.96
V1.3	18.94 ± 1.87	29.55 ± 2.12	0.84 ± 0.23	0.41 ± 0.05	0.38 ± 0.00	13.39 ± 0.81	0.253 ± 0.08	-	36.08 ± 3.10
V2.1	16.13 ± 1.19	30.52 ± 2.38	0.64 ± 0.08	0.44 ± 0.05	0.10 ± 0.01	14.00 ± 0.76	0.17 ± 0.04	-	38.11 ± 3.05
V2.2	15.45 ± 1.38	30.08 ± 2.26	0.76 ± 0.10	0.42 ± 0.04	0.18 ± 0.00	13.93 ± 0.89	0.256 ± 0.02	-	39.12 ± 3.55
V2.3	18.91 ± 1.23	31.37 ± 1.88	0.67 ± 0.07	0.37 ± 0.04	0.03 ± 0.01	14.05 ± 0.58	0.21 ± 0.03	-	34.70 ± 2.34
V3.1	15.23 ± 2.17	27.56 ± 2.97	0.79 ± 0.23	0.35 ± 0.05	0.49 ± 0.26	12.78 ± 1.28	0.186 ± 0.04	-	42.61 ± 4.28
V3.2	14.95 ± 1.93	24.33 ± 1.57	0.43 ± 0.08	0.24 ± 0.04	0.07 ± 0.01	16.67 ± 0.65	0.22 ± 0.07	0.16 ± 0.00	42.93 ± 2.08
V3.3	18.91 ± 2.34	29.20 ± 1.38	0.433 ± 0.06	0.26 ± 0.04	0.20 ± 0.08	13.99 ± 0.98	0.21 ± 0.02	-	36.80 ± 2.35
O1.1	13.75 ± 0.69	31.92 ± 1.09	0.57 ± 0.05	0.40 ± 0.02	0.04 ± 0.01	16.01 ± 0.43	0.34 ± 0.02	-	36.96 ± 1.16
O1.2	13.26 ± 2.75	36.67 ± 2.01	0.42 ± 0.07	0.19 ± 0.02	-	15.09 ± 0.89	1.20 ± 0.15	0.77 ± 0.18	31.50 ± 2.05
O1.3	13.96 ± 2.31	38.13 ± 1.40	0.73 ± 0.07	0.76 ± 0.04	0.14 ± 0.06	13.54 ± 0.05	0.45 ± 0.05	-	31.70 ± 2.31
O2.1	16.24 ± 1.60	36.06 ± 0.69	0.64 ± 0.05	0.56 ± 0.03	0.34 ± 0.30	13.92 ± 0.61	0.28 ± 0.02	-	31.95 ± 1.43
O2.2	13.75 ± 0.86	29.56 ± 1.43	0.70 ± 0.11	0.48 ± 0.04	1.12 ± 0.00	16.01 ± 0.56	0.49 ± 0.06	-	38.61 ± 1.80
O2.3	17.38 ± 1.94	29.64 ± 1.65	0.67 ± 0.07	0.48 ± 0.04	0.04 ± 0.01	14.78 ± 0.54	0.43 ± 0.13	0.33 ± 0.00	36.26 ± 1.87
O3.1	16.84 ± 1.25	31.51 ± 1.37	0.49 ± 0.045	0.36 ± 0.04	0.08 ± 0.01	14.67 ± 0.41	0.13 ± 0.02	-	35.93 ± 1.28
O3.2	21.45 ± 0.91	42.33 ± 1.76	1.45 ± 0.08	0.84 ± 0.02	0.43 ± 0.03	11.81 ± 0.83	0.25 ± 0.05	0.20 ± 0.00	21.25 ± 1.66
O3.3	20.96 ± 2.63	27.71 ± 3.64	0.62 ± 0.13	0.33 ± 0.08	0.02 ± 0.01	8.54 ± 1.26	0.22 ± 0.05	-	42.61 ± 5.8

**Table 2 materials-16-01637-t002:** Calcium mass percentages reported with EDX with their respective standard deviations.

Age Group	Species	Ca Mass Percentage (%)
Breastfed	Bovine	34.78 ± 1.63
Solid-food transitioning	Bovine	37.31 ± 2.98
Mature	Bovine	40.77 ± 2.76
Breastfed	Ovine	33.39 ± 1.84
Solid-food transitioning	Ovine	35.61 ± 1.67
Mature	Ovine	33.26 ± 2.91

**Table 3 materials-16-01637-t003:** Results of the nanoindentation tests for each evaluated sample. SFT refers to solid-food transitioning specimens.

Sample ID	Species	E (GPa)
V1.1	Breastfed Bovine	18.43 ± 1.61
V1.2	Breastfed Bovine	15.66 ± 1.46
V1.3	Breastfed Bovine	22.13 ± 2.94
V2.1	SFT Bovine	20.76 ± 3.43
V2.2	SFT Bovine	22.29 ± 1.06
V2.3	SFT Bovine	20.20 ± 3.46
V3.1	Mature Bovine	21.11 ± 2.59
V3.2	Mature Bovine	22.51 ± 1.89
V3.3	Mature Bovine	20.91 ± 2.09
O1.1	Breastfed Ovine	16.36 ± 2.25
O1.2	Breastfed Ovine	14.31 ± 2.32
O1.3	Breastfed Ovine	21.55 ± 3.95
O2.1	SFT Ovine	17.72 ± 2.99
O2.2	SFT Ovine	14.37 ± 2.57
O2.3	SFT Ovine	18.65 ± 3.17
O3.1	Mature Ovine	18.99 ± 4.21
O3.2	Mature Ovine	16.73 ± 3.81
O3.3	Mature Ovine	19.81 ± 4.10

**Table 4 materials-16-01637-t004:** Elastic modulus results presented as each age group’s means with their respective standard deviation.

Age Group	Species	E (GPa)
Breastfed	Bovine	18.74 ± 2.00
Solid-food transitioning	Bovine	21.09 ± 2.20
Mature	Bovine	21.51 ± 2.65
Breastfed	Ovine	17.40 ± 2.84
Solid-food transitioning	Ovine	16.91 ± 2.81
Mature	Ovine	18.51 ± 4.00

**Table 5 materials-16-01637-t005:** Multiple regression analysis in which all variables were regressed against the elastic modulus (R^2^ = 0.235). The bold font indicates statistically significant variables.

	Estimate	Std. Error	t Value	*p*-Value
Intercept	42.50791	4.78130	8.890	<2 × 10^−16^
Ca	−0.04145	0.07196	−0.576	0.5647
**P**	**−0.65691**	**0.11352**	**−5.787**	**9.96 × 10^−9^**
**C**	**−0.78163**	**0.12168**	**−6.424**	**2.17 × 10^−10^**
**Mature Age**	**7.82642**	**2.99406**	**2.614**	**0.0091**
**Solid-Food Transitioning Age**	**−8.77016**	**4.04234**	**−2.170**	**0.0303**
**Ovine Species**	**−8.88805**	**3.25192**	**−2.733**	**0.0064**
Ca and Mature Age	−0.11874	0.07779	−1.526	0.1272
**Ca and Solid-Food Transitioning Age**	**0.28057**	**0.11041**	**2.541**	**0.0112**
Ca and Species Ovine	0.16109	0.08588	1.876	0.0610

## Data Availability

Not applicable.

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
