# Peer review of "Variation in Juvenile Long Bone Properties as a Function of Age: Mechanical and Compositional Characterization"

_materials, 2023, doi:10.3390/ma16041637_

Round 1

Reviewer 1 Report

The study reports the mechanical properties and the chemical composition of bone with age was done for bovine and ovine animal species. The efforts take by the authors to characterise  the samples is good.

1.      The authors have identified the lack of experimental data for the human bone tissue for juvenile individuals. They have also stated the importance of mechanical characterisation will help in finite element analysis. But the reason for using the two bovine and ovine mammal species sample will be once again an approximation. The authors must give more strong reason for the selection of the specified species.

2.      The authors mention that the SEM is used for asses the is incorrect and has to be modified in section 1, page 2 and line 73-75.

3.      The authors must state clearly the suppliers of the bone with ethical clearance certificate.

4.      It has to be clearly stated that the sample are extracted from Tibia or femur because sampling is important in these experiments.

5.      The authors must state the surface roughness of all the samples as they are an important factor in this study.

6.      Is there any possibility that the usage of silicon carbide sandpaper can affect the elemental investigation of the samples.

7.      What is Poisson modulus as stated by authors in section 2 in page 4 line 152.

8.      The authors have to be more care full in preparing the samples for SEM imaging. The reasons given by authors for the identification of carbon peaks in EDX analysis are not acceptable.

9.      The figure 4 gives the nano-indentation of two samples. The authors must explain why samples V1.3 and O1.4 are given and others are not given.

10.  The typo error page 8, line 237 and page 12, line 385 has to rectified.

11.  The reason for the Elastic modulus of the ovine-SFT specimens is less than ovine- breastfed specimens has to be explained.

12.  The authors have mentioned that ovine cortical bone has reduced mechanical strength due to higher porosity page 9, line 258. The authors must provide the porosity of all the samples to validate the statement.  

13.  The authors have mentioned that the reduced sample size has influenced the inconstancy in the study on page 9, line 265 and 266. The sample size can be increased to overcome the deficiency.

14.  The discussion in page 11, line 312 to 314 a reason has to be given.

15.  The discussion in page 11, line 320 to 321 more literature evidence is requested from the authors.

16.  As mentioned by authors in page 12, line 354 to 356 more test have to be conducted to improve the quality of the paper.

17.  The author have reported that on page 12, lines 357 to 359 the experimental sites of nano-indentation and EDX are different. With respect to the above statement, how can the two results can be correlated.

18.  The authors have to do XRD analysis for the specimen.  

19.  The overall presentation has to be improved and more reference has to provided.

Author Response

The authors would like to thank the reviewers for their time and valuable comments. Please find below our responses to the questions raised. When appropriate, we have introduced the corresponding changes in the manuscript using the track changes tool in word. We have updated two manuscript versions: one of them with tracked changes and another one with the aforementioned modifications already accepted.

  1. The authors have identified the lack of experimental data for the human bone tissue for juvenile individuals. They have also stated the importance of mechanical characterisation will help in finite element analysis. But the reason for using the two bovine and ovine mammal species sample will be once again an approximation. The authors must give more strong reason for the selection of the specified species.

Thanks for your comment. We have clarified this point in the new version of the manuscript. This study is a first step towards obtaining reliable material descriptions of pediatric tissue, which is the end goal. The main focus of this study was to confirm that similar trends in the change of material properties could be observed in two different species of mammals. If this is true and this trend can be modelled, perhaps a similar trend can also be proposed for the human. In the case of the human species, we could scale from adults to pediatric subjects. However, again, this study focused only on identifying whether two different mammal species (that were chosen for convenience) showed similar trends in change in material properties with development.

  1. The authors mention that the SEM is used for asses the is incorrect and has to be modified in section 1, page 2 and line 73-75.

Corrected as suggested.

  1. The authors must state clearly the suppliers of the bone with ethical clearance certificate.

All samples were obtained from the grocery store and were neither grown nor sacrificed for the purpose of our study. This is why we did not seek any kind of ethical approval, as it is not needed in this case.

  1. It has to be clearly stated that the sample are extracted from Tibia or femur because sampling is important in these experiment.

All the bovine samples are from femur, but unfortunately, we do not have records for the ovine to state if they come from femur or tibia. That is why we mention that bone wafers were prepared from both femurs and tibias in the manuscript.

  1. The authors must state the surface roughness of all the samples as they are an important factor in this study.

The authors agree on the importance of surface roughness in the accurate measurement of nano-hardness, for this purpose, embedded samples where polished to 1 µm diamond until pristine surfaces where achieved. This was assessed by optical microscopy visual inspection. However, we do not have quantitative data of samples surface roughness.

  1. Is there any possibility that the usage of silicon carbide sandpaper can affect the elemental investigation of the samples.

No, the EDX elemental investigations were performed on as received and cleaned samples. After EDX examinations, samples were embedded, polished, and later tested in the nanoindenter, therefore, no contamination is expected.

However, before metallographic preparation, the authors carefully investigated the most suitable methods for the preparation of bones specimens for nanoindentation. In this regard, the following authors reported the same metallographic preparation steps with successful results:

Casanova M, Balmelli A, Carnelli D, Courty D, Schneider P, Müller R. 2017 Nanoindentation analysis of the micromechanical anisotropy in mouse cortical bone.R. Soc. open sci. 4: 160971. http://dx.doi.org/10.1098/rsos.160971

Huja, S.S., Beck, F.M. & Thurman, D.T. Indentation Properties of Young and Old Osteons. Calcif Tissue Int 78, 392–397 (2006). https://doi.org/10.1007/s00223-006-0025-3

  1. What is Poisson modulus as stated by authors in section 2 in page 4 line 152.

The Poisson’s modulus/ratio has not been calculated in our research, a value of 0.3 has been assumed after performing an intensive literature review to identify the mechanical properties of cortical bone.

  1. The authors have to be more careful in preparing the samples for SEM imaging. The reasons given by authors for the identification of carbon peaks in EDX analysis are not acceptable.

The authors agree on the importance of carefully selecting the appropriate metallographic procedure for SEM imaging. In this regard, the nonconductive nature of our specimens made it necessary to coat them with carbon, which was the only material available in our lab for such a purpose. Nevertheless, our carbon coater allows for the deposition of a set thickness, so all specimens were coated under the same conditions; thus, comparisons are possible. Furthermore, the carbon peak of the X-ray graph does not interfere with other peaks of other elements. 

  1. The figure 4 gives the nano-indentation of two samples. The authors must explain why samples V1.3 and O1.4 are given and others are not given.

We have chosen these two samples just for illustration purposes, but all the obtained nanoindentation curves showed similar behavior, this fact has been clarified in the text. Nevertheless, the plot below is showing the curves obtained for other of the specimens involved in the study.

  1. The typo error page 8, line 237 and page 12, line 385 has to rectified.

Thank you very much for finding typo. Corrected as suggested.

  1. The reason for the Elastic modulus of the ovine-SFT specimens is less than ovine- breastfed specimens has to be explained.

In reality, it is not lower as there were not statistically significant differences between the two groups. This is consistent with previous literature (see references [1,2]). But it is true that we were expecting a higher value regardless of the significance. We have added extra text to the discussion. Interestingly the regression model found that the increase in the mature specimens was statistically significant.

  1. The authors have mentioned that ovine cortical bone has reduced mechanical strength due to higher porosity page 9, line 258. The authors must provide the porosity of all the samples to validate the statement.  

The authors agree that our findings will benefit from quantitative data regarding porosity. Unfortunately, we do not have these data and it is not possible for us to perform these measurements right now. In this regard, the only evidence we can provide at the moment are the SEM pictures of a parallel study that is being carried out with bovine and ovine specimens obtained from the same tibias and femurs that we have used in this research. In those images (included at the end of the discussion), it can be clearly seen that the cortical bone of ovine is more porous than bovine, however, no quantitative data, has been performed yet

  1. The authors have mentioned that the reduced sample size has influenced the inconstancy in the study on page 9, line 265 and 266. The sample size can be increased to overcome the deficiency.

We believe we will continue with this work in the future, but at the moment we do not have the opportunity to include a larger sample size to the current study. However, we have detected significant results with the current sample population.

  1. The discussion in page 11, line 312 to 314 a reason has to be given.

We did not feel comfortable with these sentence without making a quantitively comparison of the C content, so we decided to remove these lines from the manuscript.

  1. The discussion in page 11, line 320 to 321 more literature evidence is requested from the authors.

References included as suggested.

  1. As mentioned by authors in page 12, line 354 to 356 more test have to be conducted to improve the quality of the paper.

Yes, in fact, there are plans to increase the sample size and therefore testing. The results included in the manuscript are a pilot study carried out to confirm the hypothesis and whether the resolution of the techniques used was sufficient to detect differences between the developmental stages of the tissue. We hope that the publication of these results will support the continuation of this study to increase the sample population.

  1. The author have reported that on page 12, lines 357 to 359 the experimental sites of nano-indentation and EDX are different. With respect to the above statement, how can the two results can be correlated.

The results can be correlated because the specimens tested via EDX and nanoindentation are the same. It is true that the experimental sites many not be the same, but a point-cloud approximation throughout the whole surface of the samples was used in a way to obtain representative data of each material.

  1. The authors have to do XRD analysis for the specimen.

XRD analysis is definitely a technique that could help to enhance the evidence of our findings. Unfortunately, we do not have access to this equipment. However, providing further information about phases, grain size, texture, % crystallinity and stress state, would have been useless in amorphous or low crystallinity phases such as bone apatite due to the presence of organic polymers, as indicated by [1].

On the contrary, EDX presented several benefits for this investigation, such as the fact that it required less information about the sample’s compositions, which were, a priori, unknown, and the possibility of getting area-specific elemental analysis, avoiding porous or nonuniform areas of the material.

[1] Rogers KD, Daniels P. An X-ray diffraction study of the effects of heat treatment on bone mineral microstructure. Biomaterials. 2002 Jun;23(12):2577–2585

  1. The overall presentation has to be improved and more reference has to provided.

As gently suggested by the Reviewer, we have re-written this section including results and conclusions obtained after the study. 

Reviewer 2 Report

The paper entitled “Juvenile long bone properties variation as a function of age: mechanical and compositional characterization” presents the characterisation of cortical bone using nanoindentation and spectroscopy. The paper is well written and offers excellent data that deserved to be published. The data will be especially useful in the development of personalised tissue engineering bone scaffold.

1.       There are areas where it says reference missing. This is likely to be a formatting error. It is recommended that the authors check for formatting errors throughout the manuscript.

2.       The paper will benefit from a conclusion section summarising the key findings.

3.       The observations of elastic modulus are consistent with that of previously reported, therefore discussion would benefit from making few comparisons.

4.       The introduction will also benefit from reinforcing the importance of this study to bone scaffold, some relevant literature may also be cited.

Classification of Biomaterial Functionality. Encyclopedia of Smart Materials, Volume 1, 2022, Pages 86-102.

The Mechanical Properties of Bone Tissue Engineering Scaffold Fabricating Via Selective Laser Sintering. Life System Modeling and Simulation. LSMS 2007. Lecture Notes in Computer Science, vol 4689. Springer, Berlin.

5.       What are the limitations and prospects of this study?

Author Response

The authors would like to thank the reviewers for their time and valuable comments. Please find below our responses to the questions raised. When appropriate, we have introduced the corresponding changes in the manuscript using the track changes tool in word. We have updated two manuscript versions: one of them with tracked changes and another one with the aforementioned modifications already accepted.

The paper entitled “Juvenile long bone properties variation as a function of age: mechanical and compositional characterization” presents the characterisation of cortical bone using nanoindentation and spectroscopy. The paper is well written and offers excellent data that deserved to be published. The data will be especially useful in the development of personalised tissue engineering bone scaffold.

  1. There are areas where it says reference missing. This is likely to be a formatting error. It is recommended that the authors check for formatting errors throughout the manuscript.

Thank you very much for finding typo. Done as suggested.

  1. The paper will benefit from a conclusion section summarising the key findings.

Modified as suggested.

  1. The observations of elastic modulus are consistent with that of previously reported, therefore discussion would benefit from making few comparisons.

Included as suggested.

  1. The introduction will also benefit from reinforcing the importance of this study to bone scaffold, some relevant literature may also be cited.

Classification of Biomaterial Functionality. Encyclopedia of Smart Materials, Volume 1, 2022, Pages 86-102.

The Mechanical Properties of Bone Tissue Engineering Scaffold Fabricating Via Selective Laser Sintering. Life System Modeling and Simulation. LSMS 2007. Lecture Notes in Computer Science, vol 4689. Springer, Berlin.

Included as suggested.

Reviewer 3 Report

The authors are reporting methods that can be used to quantify the change in mechanical properties and composition at different stages of bone tissue development.

·    The authors stated that nonparametric variance tests revealed that there were no statistically significant Ca content variations within the different age groups, the compositional analysis revealed that calcium mass percentage, and therefore mineralization and crystallization, increased with age for juvenile tissue in both species.

What are the causes of Ca compositional changes with age (mechanism)? And how does it affect the structure-mechanical properties of the bone?

·        The discussion of the study is very concerning because the majority is just literature reports (references) and a small portion discusses the findings. This raises questions about the novelty and contribution of the findings in advancing the field of material characterization. Can the authors please comment on that?

·        The authors should look at the paper (ref 4) that was published in 2016 by Mohammad J. Mirzaali et al. Tittle “Mechanical properties of cortical bone and their relationships with age, gender, composition and microindentation properties in the elderly” doi: 10.1016/j.bone.2015.11.018.

“In that paper, it was reported that age, age, with minor exceptions gender, and small variations in average mineralisation have a negligible effect on the tissue microindentation properties of human lamellar bone in the elderly. Furthermore, their findings suggest that microindentation experiments are suitable to predict macroscopic mechanical properties in the elderly only on average and not on a one to one basis. The presented data may help to better understand the mechanisms of ageing in bone tissue and the length scale at which they are active.”

Therefore, I feel that there’s a redundancy between that study and what is being reported here. The authors should clarify the difference between the two and highlight their work’s novelty and possible impact.

·        Lastly, I would like to know if any plans are there to do the study in a larger population for statistical purposes.

Author Response

The authors would like to thank the reviewers for their time and valuable comments. Please find below our responses to the questions raised. When appropriate, we have introduced the corresponding changes in the manuscript using the track changes tool in word. We have updated two manuscript versions: one of them with tracked changes and another one with the aforementioned modifications already accepted.

1.The authors are reporting methods that can be used to quantify the change in mechanical properties and composition at different stages of bone tissue development. The authors stated that nonparametric variance tests revealed that there were no statistically significant Ca content variations within the different age groups, the compositional analysis revealed that calcium mass percentage, and therefore mineralization and crystallization, increased with age for juvenile tissue in both species. What are the causes of Ca compositional changes with age (mechanism)? And how does it affect the structure-mechanical properties of the bone?

Bone development usually experiences remarkable changes regarding the shape and size of the tissue. Bone growth is ruled by the continuous Ca homeostatic remodeling, which repairs fractures and gives shape to the bone during the early stages (1). Calcium and mineral percentages in bone are different for each individual, and they may be related to genetic factors (2). However, Ca bone quantity should be constant, and this is achieved with a balanced nutrition. There are references (already included in the manuscript and noted below) in the literature which show that Ca percentages remain almost constant in juvenile subjects right before puberty, and this could be one of the reasons why we do not find statistically significances within different age groups (3). This explanation has been added to the manuscript in discussion part as well.

(1) Nap RC, Hazewinkel HA. Growth and skeletal development in the dog in relation to nutrition; a review. Vet Q. 1994; 16:50-9

(2) Ferrari, Rizzoli R, Manen D, Slosman D, Bonjour JP. Vitamin D receptor gene start codon polymorphisms (FokI) and bone mineral density: interaction with age, dietary calcium, and 3'-end region polymorphisms. J Bone Miner Res. 1998; 13:925-30

(3) Bonjour JP, Chevalley T, Ferrari S, Rizzoli R. The importance and relevance of peak bone mass in the prevalence of osteoporosis. Salud Publica Mex. 2009; 51(Suppl 1):S5-S17

The higher the Ca content, the higher the stiffness should be, or in other words, higher Elastic modulus values will be reported.

  1. The discussion of the study is very concerning because the majority is just literature reports (references) and a small portion discusses the findings. This raises questions about the novelty and contribution of the findings in advancing the field of material characterization. Can the authors please comment on that?

Thanks for your comment. As other reviewer had suggested, we have highlighted the novelty of the study within the conclusions section. There is a clear contribution of this paper that is to provide experimental data from juvenile specimens. Given the lack of mechanical testing performed on developing tissue, adding more data to the body of the literature in the field can be considered a contribution in itself.

Secondly, this is the first study, to our knowledge, that compares directly the mechanical behavior and composition of specimens from two different species and with exactly the same methods. This methodology allowed us to identify similar trends in the mechanical behavior of the two species.

  1. The authors should look at the paper (ref 4) that was published in 2016 by Mohammad J. Mirzaali et al. Tittle “Mechanical properties of cortical bone and their relationships with age, gender, composition and microindentation properties in the elderly” doi: 10.1016/j.bone.2015.11.018.

“In that paper, it was reported that age, age, with minor exceptions gender, and small variations in average mineralisation have a negligible effect on the tissue microindentation properties of human lamellar bone in the elderly. Furthermore, their findings suggest that microindentation experiments are suitable to predict macroscopic mechanical properties in the elderly only on average and not on a one to one basis. The presented data may help to better understand the mechanisms of ageing in bone tissue and the length scale at which they are active.”

Therefore, I feel that there’s a redundancy between that study and what is being reported here. The authors should clarify the difference between the two and highlight their work’s novelty and possible impact.

The aforementioned paper focused on elderly subjects, while our research studies the mechanical properties and composition of in-development juvenile bone tissue. As mentioned above, given the scarcity of experimental data on pediatric tissue in the literature, we believe that the data included in the article can be useful to the scientific community.

  1. Lastly, I would like to know if any plans are there to do the study in a larger population for statistical purposes.

Yes, there are plans to increase the sample size. The results included in the manuscript are a pilot study carried out to confirm the hypothesis and whether the resolution of the techniques used was enough to detect the differences between developmental stages of the tissue. We hope that the publication of these results will support the continuation of this study to increment the sample population.

Reviewer 4 Report

This study investigated the variations of material component and mechanical properties in juvenile bovine and ovine bones with the development of age. However, the method and results descriptions are unprecise and incomplete, so the data is not convinced enough. And there seems to be no novel conclusions or ideas. This study is mainly about the detection of phenomena. What exactly scientific problems does it want to explain?

Method and materials:

1. Paragraph 1: do the “mature individuals” in the study have the upper limit of age for two types of animals?

2. Paragraph 3: the sentence “Eighteen bone wafers were machined for this study … were made for each group” is confused, didn’t the authors classify the specimens by the animal type? Or the samples from different types of animals were mixed together to test? According to the statistical analysis described, the author considered about the difference of species, so the grouping needs to be reconsidered.

3. Paragraph 5: I doubt the author can use SEM to identify the phase composition of mineral bone. The device that can distinguish the variations of compositions is X-ray dispersion detectors. The author should modify the statement.

4. Paragraph 6: doesn’t metalizing process have some effect on the X-ray dispersion of the samples? The final energy dispersive X-ray of bone materials may be changed after metalizing.

5. Paragraph 7: after SEM observation, did the author do nanoindentation on the metalized samples? If so, the tested elastic modulus of samples will not be the actual value and the results will not be convinced.

6. How to do X-ray analysis? Did author test X-ray dispersion of one point or one field on samples? The method should be described.

7. Statistical analysis: what’s the description of significant differences between groups?

Results: Generally speaking, the statements of this part is not rigorous and objective, such as the differences of main components between groups should be showed on figures, statistical results do not need to be described independently, and the author should discuss the results in Discussion part but not here.

3.1:

1. Why the author can detect sulfur on undecalcified bones? In which four samples? Did they follow some rules?

2. Table 1: the exhibition of the data is incorrect. The author should show the data by mean value with standard diviation for each group. 

3. As Figure 2 shows, the samples were divided into 6 groups but not 3 groups mentioned in method part. So the statistical analysis should use two-way ANOVA instead of one-way variance analyses.

4. Line200-204: the statements are more like discussion but not the results, since the author did not quantify these components.

5. Line205-209: how did the author know carbon content did not change without precise data between groups?

3.2:

1. Table 2 has the same problem with table 1.

2. The author only showed the elastic modulus of samples, but how about the hardness?

3. Line 237-238: “…shown in Error! Reference source not found..”.

3.3:

1. How to interpret the data in table 3? What do these indicators mean lay in the first row? What’s more, the author should describe the results but not discuss it in Result part.

Discussion:

1. Line 306-308: “…the compositional analysis revealed that calcium mass percentage…”, the sentence is incomplete.

2. Line 346-347: “Kruskal-Wallis tests could not find statistically significant changes in stiffness with age within species…” Did author tested the stiffness of samples?

3. Line 349-350: “In parallel, an interaction between the Ca content and the solid food transitioning group was also identified in the regression model”. From the regression description in Results part, I cannot obtain this conclusion.

4. Line 361-371: what is the relationship of this paragraph with this work?

5. Line 381: …higher porosity content that is typical for ovine bone tissue.” The author has been emphasized the higher porosity in ovine bone than in bovine bone, but how to prove it?

6. Line 385: “(Error! Reference source not found.).”

Author Response

The authors would like to thank the reviewers for their time and valuable comments. Please find below our responses to the questions raised. When appropriate, we have introduced the corresponding changes in the manuscript using the track changes tool in word. We have updated two manuscript versions: one of them with tracked changes and another one with the aforementioned modifications already accepted.

This study investigated the variations of material component and mechanical properties in juvenile bovine and ovine bones with the development of age. However, the method and results descriptions are unprecise and incomplete, so the data is not convinced enough. And there seems to be no novel conclusions or ideas. This study is mainly about the detection of phenomena. What exactly scientific problems does it want to explain?

Thank you for your comments. There is a clear and novel contribution of this article, that is, providing experimental data from juvenile specimens. Given the lack of mechanical tests performed on developing tissue, adding more data to the body of literature in the field can be considered a contribution in itself.

Secondly, this is the first study, to our knowledge, that directly compares the mechanical behavior and composition of specimens from two different species and with exactly the same methods. This methodology allowed us to identify similar trends in the mechanical behavior of the two species. Furthermore, the results included in the manuscript are a pilot study carried out to confirm the hypothesis and whether the resolution of the techniques used was sufficient to detect the differences between the developmental stages of the tissue. We hope that the publication of these results will support the continuation of this study to increase the sample population and to check if these outputs could be modelled to understand how bone tissue behaves in humans, scaling results from adults to children.

Method and materials:

  1. Paragraph 1: do the “mature individuals” in the study have the upper limit of age for two types of animals?

No. Please note that we could not control exactly the age of the specimens and we used the transitioning to solid food as a proxy for age, as this transition depends on the age of the animal. These age groups are listed in the first paragraph of the Materials and Methods section.

  1. Paragraph 3: the sentence “Eighteen bone wafers were machined for this study … were made for each group” is confused, didn’t the authors classify the specimens by the animal type? Or the samples from different types of animals were mixed together to test? According to the statistical analysis described, the author considered about the difference of species, so the grouping needs to be reconsidered.

In fact, samples were classified with respect to the species and developmental stage. As mentioned, 18 bone wafers were machined for this study. The samples were classified into three different groups (breastfed, solid-food transitioning and mature) and six samples were made for each group (one for each age and species, that is, group one contained a breastfed bovine sample, a solid-food transitioning bovine sample, a mature bovine sample, a breastfed ovine sample, a solid-food transitioning ovine sample, and a mature ovine sample). Each group contained both species, but this did not alter the testing. I hope the aforementioned helped to understand the grouping.

  1. Paragraph 5: I doubt the author can use SEM to identify the phase composition of mineral bone. The device that can distinguish the variations of compositions is X-ray dispersion detectors. The author should modify the statement.

Corrected as suggested. The technique used to obtain compositional information of the samples is Energy Dispersive X-ray analysis, which is performed using the scanning electron microscope.

  1. Paragraph 6: doesn’t metalizing process have some effect on the X-ray dispersion of the samples? The final energy dispersive X-ray of bone materials may be changed after metalizing.

The metallizing process deposits a nanometric Carbon layer on the sample’s surface due to the non-conductive nature of our specimens. It is performed on an inert atmosphere at RT, so the unique changes it may confer to the X-ray dispersion results are those related to the intensity of the carbon peaks, as stated in the manuscript. Nevertheless, our carbon coater allows for the deposition of a set thickness, so all specimens were coated under the same conditions; thus, comparisons are possible. Furthermore, the carbon peak of the X-ray graph does not interfere with other peaks of other elements.

  1. Paragraph 7: after SEM observation, did the author do nanoindentation on the metalized samples? If so, the tested elastic modulus of samples will not be the actual value and the results will not be convinced.

We agree on the fact that a coating on sample’s surfaces could interfere in the elastic properties of materials, since the penetration of the indenter is of ~600 nm. However, it should be noted that samples were coated and observed under the SEM before metallographic preparation and indentation analysis, so no interference is possible, as the nanometric carbon layer was completely removed before measuring specimens’ elastic modulus.

  1. How to do X-ray analysis? Did author test X-ray dispersion of one point or one field on samples? The method should be described.

Energy dispersive X-ray analysis has been performed using a point cloud matrix throughout the whole surface of the samples in order to get average representative results of the specimens’ composition. This information has been included in the manuscript (lines 127-129), as suggested by the reviewer.

  1. Statistical analysis: what’s the description of significant differences between groups?

We considered that the results were significant whenever the p-value reached a value smaller than 0.05. It is included in the manuscript.

Results: Generally speaking, the statements of this part is not rigorous and objective, such as the differences of main components between groups should be showed on figures, statistical results do not need to be described independently, and the author should discuss the results in Discussion part but not here.

As gently suggested by the Reviewer, we have modified these sections.

3.1:

2

  1. Why the author can detect sulfur on undecalcified bones? In which four samples? Did they follow some rules?

Sulphur is a constituent of bones, especially present in collagen. It is usually found in tissues that contain large amounts of protein. Small proportions of Sulphur were detected in our results, and they did not follow any clear rule.

  1. Table 1: the exhibition of the data is incorrect. The author should show the data by mean value with standard diviation for each group.

As gently suggested by the reviewer, this information has been included in the text.

  1. As Figure 2 shows, the samples were divided into 6 groups but not 3 groups mentioned in method part. So the statistical analysis should use two-way ANOVA instead of one-way variance analyses.

By including Figure 2 and Figure 5 we wanted to depict the variation of Ca and elastic modulus for each age group, respectively. Regarding the statistical analyses, the effect of species (bovine and ovine) on composition and elastic modulus was analyzed. After checking the homogeneity of the variance, Mann Whitney U tests were performed to compare the statistical significance of the two species in calcium content and elastic modulus. Since we were comparing the variation of these parameters (first for calcium and second for elastic modulus) between two species (two groups), one-way variance tests were used.

  1. Line200-204: the statements are more like discussion but not the results, since the author did not quantify these components.

Modified as suggested.

  1. Line205-209: how did the author know carbon content did not change without precise data between groups?

Prior to EDX analysis, all samples were carbon-coated under the same conditions, with the same thickness, thus comparisons between the elemental composition of each specimen are possible.

  1. Table 2 has the same problem with table 1.

Included as suggested.

  1. The author only showed the elastic modulus of samples, but how about the hardness?

Unfortunately, we did not include data regarding hardness of the samples because it is out of the scope of this study.

  1. Line 237-238: “…shown in Error! Reference source not found.”.

Corrected as suggested.

3.3:

  1. How to interpret the data in table 3? What do these indicators mean lay in the first row? What’s more, the author should describe the results but not discuss it in Result part.

The results in Table 3 (now Table 5 in the modified manuscript) correspond to the coefficients, corresponding standard errors, and significance (p-value) of the regression model used to estimate the relationship of the modulus with the different variables included in the model. The explanation of the coefficients that were significant in predicting the value of the modulus of elasticity has been provided in the two paragraphs above the table (Page 10, line 294 (v1 version) or 309  (tracked changes document). We have edited the text to make the information contained in the table clearer.

Discussion:

  1. Line 306-308: “…the compositional analysis revealed that calcium mass percentage…”, the sentence is incomplete.

Sentence has been completed.

  1. Line 346-347: “Kruskal-Wallis tests could not find statistically significant changes in stiffness with age within species…” Did author tested the stiffness of samples?

Yes, as depicted in Section 2 (lines 138 to 170), stiffness was tested by using the nanoindentation technique to determine the elastic modulus, the results obtained can be observed in Table 3.

  1. Line 349-350: “In parallel, an interaction between the Ca content and the solid food transitioning group was also identified in the regression model”. From the regression description in Results part, I cannot obtain this conclusion.

The interaction between the Ca content and the solid-food transitioning samples is given by one of the independent variables included in the model, which was formulated as the product of the Ca concentration and the dummy variable identifying the SFT group. This variable resulted statistically significant (p=0.0112) and therefore we conclude that there is an interaction between these two values.

  1. Line 361-371: what is the relationship of this paragraph with this work?

This paragraph has been included to compare our results with previous literature references that support our findings. Specifically, the topic of this paragraph is the relationship between bone tissue stiffness and calcium content for different group ages.

3

  1. Line 381: “…higher porosity content that is typical for ovine bone tissue.” The author has been emphasized the higher porosity in ovine bone than in bovine bone, but how to prove it?

Unfortunately, we cannot provide quantitative data regarding porosity because we have not measured it. The only evidence we can provide right now are the SEM pictures of a parallel study that is being carried out with bovine and ovine samples (images included at the end of the discussion part) obtained from the same tibias and femurs that we have used, in which it can be clearly seen that ovine cortical bone is more porous than bovine. We do not have any quantitatively data and it is not possible for us to perform these measurements right now.

  1. Line 385: “(Error! Reference source not found.

Corrected as suggested.

Round 2

Reviewer 1 Report

corrections to minor methodological errors and text editing.

Author Response

Thanks again for your review of our work, we appreciate your time.We have edited again the manuscript to correct potential misuses, as suggested. We hope that if there are any edits left, the editorial team will catch them and correct them. Thank you.

Reviewer 3 Report

The authors were able to address all the concerns I had with the manuscript. Barring minor spelling/grammar corrections, the manuscript is recommended for publication. 

Author Response

Thank you again for your review of our work, we appreciate your time.

Reviewer 4 Report

According to the revised version, I suggest the manuscript needs to be modified again before acceptance consideration. The author answer the majority parts of the questions I proposed last time, while still remains some need to be solved.

1.       The authors keep emphasizing the mechanical properties test for bone, however, the test of elastic modulus by nanoindentation on bone is unilateral, which only reflect the stiffness and elastic deformation on bone surface. When nanoindentation was conducted, the tips of machine inevitably remained scratch on bone surface, making a plastic deformation occurred. Therefore the hardness of bone also needs to be determined by nanoindentation. Hardness is a comprehensive mechanical property index, which also reflect the strength of material.

2.       I can’t understand why the author exhibited all the data of each individual sample (Table 1 and Table 3). Is that necessary? Table 2 and Table 4 is clear enough after grouped, only need to mark the significantly difference on data.

3.       The comparison on two species about Ca content and elastic modulus showed in Section 3.3. Did the author do the comparison between two species on the same age stage, or on the all ages for each specie? The author needs to marked stars on Figure 2 and Figure 5 to show the significantly differences between two species. 

4.       Why only “ovine species” showed in Table 5 without “bovine”? The interpretation of multiple regression model in Results part was not clarity enough. What’s mean by “Estimate” and “t-value” for each variable in Table 5?

5.       English gramma needs to be improve, such as in Line 424-425: ”… the breastfed ovine specimens exhibited the same elastic modulus than the SFT ovine specimens…”, it should be “with” but not “than”.

Author Response

According to the revised version, I suggest the manuscript needs to be modified again before acceptance consideration. The author answers the majority parts of the questions I proposed last time, while still remains some need to be solved.

Thanks again for your review of our work. We appreciate your time and hope that our responses below (in blue font) help to sort out the remaining issues.

1.       The authors keep emphasizing the mechanical properties test for bone, however, the test of elastic modulus by nanoindentation on bone is unilateral, which only reflect the stiffness and elastic deformation on bone surface. When nanoindentation was conducted, the tips of machine inevitably remained scratch on bone surface, making a plastic deformation occurred. Therefore the hardness of bone also needs to be determined by nanoindentation. Hardness is a comprehensive mechanical property index, which also reflect the strength of material.

The authors agree that hardness is a relevant mechanical property, and therefore we would like to include values of this parameter in the future continuation of the project. It is true that we currently have these values for the as-studied samples, since the nanoindenter provides hardness as an output. However, the area affected by the indenter tip has major elastic contributions rather than plastic, so it will be not enough to simply include these hardness values. Imprints should be carefully analyzed in order to check if phenomena such as plie-out, sink-in or crack generation happened and affected hardness results. So, we believe we could include a more detailed study regarding this parameter in further research, such as the one performed in the following reference:

Wu WW, Zhu YB, Chen W, Li S, Yin B, Wang JZ, Zhang XJ, Liu GB, Hu ZS, Zhang YZ. Bone Hardness of Different Anatomical Regions of Human Radius and its Impact on the Pullout Strength of Screws. Orthop Surg. 2019 Apr;11(2):270-276. doi: 10.1111/os.12436. Epub 2019 Mar 25. PMID: 30908880; PMCID: PMC6594527.

In the new version of the manuscript, we have rephrased the objective of the paper to reflect that we are only focusing on the change in the elastic modulus of the tissue.

  1. I can’t understand why the author exhibited all the data of each individual sample (Table 1 and Table 3). Is that necessary? Table 2 and Table 4 is clear enough after grouped, only need to mark the significantly difference on data.

We would like to keep these two tables as the information included is necessary for anyone trying to replicate the study. Since there is no limitation about the number of tables in the journal, we would prefer to keep them as they are.

  1. The comparison on two species about Ca content and elastic modulus showed in Section 3.3. Did the author do the comparison between two species on the same age stage, or on the all ages for each specie? The author needs to marked stars on Figure 2 and Figure 5 to show the significantly differences between two species ¿?.

Section 3.3 describes the results obtained from the statistical tests. Two different tests were performed:

             - Mann Whitney U: these tests were carried out first in order to determine if any statistical difference existed within species regarding Ca content, and secondly in order to check if there were statistical differences between species when considering elastic modulus, in both cases regardless of age.

             -Kruskal Wallis: these tests were performed first so as to evaluate if any statistical difference existed within the three different age groups regarding Ca content, and secondly to determine if there were statistical differences between the three different age groups regarding the elastic modulus.

I hope this explanation can clarify your doubts.

  1. Why only “ovine species” showed in Table 5 without “bovine”? The interpretation of multiple regression model in Results part was not clarity enough. What’s mean by “Estimate” and “t-value” for each variable in Table 5?

When you include categorical variables in a regression model, one of the categories in the categorical variable is taken as the baseline or reference level. This is how the method works. The meaning is that -since this variable is significant (p-value<0.05), coupons from ovine species are essentially different from coupons of bovine species. The t-value is the value of the T-Student distribution that allows to calculate the p-value that gives the significance of the coefficient. Estimate is the estimation of the coefficient of the regression model that allows to calculate the value of the dependent value as the combination of the independent variables. All these concepts are inherent to the development of a regression model and can be found in any text about this statistical approach. We have added a reference in case the readers would like to understand better the method (Bland, 2015). Thank you.

5.       English gramma needs to be improve, such as in Line 424-425: ”… the breastfed ovine specimens exhibited the same elastic modulus than the SFT ovine specimens…”, it should be “with” but not “than”.

We have edited again the manuscript to correct potential misuses, as suggested. We hope that if there are any edits left, the editorial team will catch them and correct them. Thank you.